# Sex- and strain-dependent effects of ageing on sleep and activity patterns in *Drosophila*

**Nathan Woodling** [ORCID]*

School of Molecular Biosciences, University of Glasgow, Glasgow, United Kingdom

* nathan.woodling@glasgow.ac.uk

**Data Availability Statement:** All raw data and metadata underlying each figure can be freely obtained at Mendeley Data, https://doi.org/10. 17632/8633hm46p5.1.

## Abstract

The fruit fly *Drosophila* is a major discovery platform in the biology of ageing due to its balance of relatively short lifespan and relatively complex physiology and behaviour. Previous studies have suggested that some important phenotypes of ageing, for instance increasingly fragmented sleep, are shared from humans to *Drosophila* and can be useful measures of behavioural change with age: these phenotypes therefore hold potential as readouts of healthy ageing for genetic or pharmacological interventions aimed at the underpinning biology of ageing. However, some age-related phenotypes in *Drosophila* show differing results among studies, leading to questions regarding the source of discrepancies among experiments. In this study, I have tested females and males from three common laboratory strains of *Drosophila* to determine the extent to which sex and background strain influence age-related behavioural changes in sleep and activity patterns. Surprisingly, I find that some phenotypes–including age-related changes in total activity, total sleep, and sleep fragmentation–depend strongly on sex and strain, to the extent that some phenotypes show opposing age-related changes in different sexes or strains. Conversely, I identify other phenotypes, including age-related decreases in morning and evening anticipation, that are more uniform across sexes and strains. These results reinforce the importance of controlling for background strain in both behavioural and ageing experiments, and they imply that caution should be used when drawing conclusions from studies on a single sex or strain of *Drosophila*. At the same time, these findings also offer suggestions for behavioural measures that merit further investigation as potentially more consistent phenotypes of ageing.

## Introduction

The fruit fly *Drosophila* is one of the most widely used model organisms in ageing research: its relatively short lifespan, low maintenance cost, and extensive genetic tractability have allowed *Drosophila* research to complement work in *C. elegans* and other organisms in discovering a number of highly evolutionarily conserved genes and signalling pathways whose genetic and/ or pharmacological activation or inhibition can extend lifespan [1]. Importantly, in many cases the lifespan-extending effects of these interventions are highly conserved from less complex species like *C. elegans* to more complex animals like mice [2–4].

**Funding:** This study was funded by the Royal Society (https://royalsociety.org/), Research Grant RGS\R1\231248 to NW. The funders played no role in study design, data collection and analysis, decisions to publish, or preparation of the manuscript.

**Competing interests:** The author has declared that no competing interests exist.

In addition to lifespan, healthspan (the amount of life spent in good health) is arguably an even more important outcome to consider in ageing research [5]. *Drosophila* have offered powerful tools for studying healthspan, with a prominent example in the study of sleep quality changes with age. In flies [6,7], mice [8], and humans [9–11], sleep is reported to become more fragmented with age, as measured by shorter individual sleep episodes or bouts. Importantly, fragmented sleep appears to be an important correlate of reduced healthspan in humans, as sleep fragmentation is predictive of both dementia and all-cause cognitive decline [12–14]. *Drosophila* have offered additional evidence for correlations between sleep patterns and lifespan: for instance, indices of sleep fragmentation and circadian rhythmicity can significantly predict lifespan for individual flies, albeit with relatively low predictive power [15]. *Drosophila* studies have also shown that lifespan-extending interventions can reduce age-related sleep fragmentation: for instance, deletion of genes encoding insulin-like peptides or treatment with the TOR inhibitor rapamycin can both extend lifespan [16,17] and reduce age-related sleep fragmentation in flies [18]; similar effects of increased lifespan and reduced sleep fragmentation have been observed for inhibition of the receptor tyrosine kinase Alk in neurons [19]. Taken together, these findings suggest that age-related changes in sleep and activity patterns can be one useful behavioural measure of healthspan in *Drosophila*.

At the same time, several important caveats exist for these studies. Firstly, many studies, including those discussed above [16,17,19], have shown sex-specific or sex-biased effects of lifespan-extending interventions in *Drosophila* [20], suggesting that some biological mechanisms of ageing may be sex-specific. Secondly, many studies of *Drosophila* ageing are done within a single laboratory strain as a common genetic background–a practice that allows for careful control of confounding factors from genetic background, but also a practice that often limits studies to a single background strain and may thus limit the universality of findings. A careful analysis of behavioural ageing among strains and between sexes will therefore be a useful starting point from which future studies on *Drosophila* lifespan and healthspan can build. To that end, I have here investigated whether several commonly used measures of sleep and activity patterns show similar age-related trajectories in female and male flies from three different laboratory genetic background strains. My results show a surprising degree of divergence among sexes and strains for many age-related phenotypes, while highlighting some more consistent age-related behavioural changes that may guide future studies.

## Results

### Lifespan diverges among *Drosophila* strains and sexes

To begin investigating differences in lifespan and behaviour among strains and sexes, I chose three genetic background stocks (hereafter 'strains') of *Drosophila* commonly used in studies of sleep and/or ageing (see **Material and Methods**): *Canton-S* (*CS*), a wild-type (red-eyed) stock commonly used in behaviour research; *Dahomey* (*Dah*), a wild-type (red-eyed) population commonly used in ageing research; and a lab stock containing a mutant *white* allele ($w^{1118}$), a stock commonly used as a background for genetic experiments involving transgenes marked by the presence of a 'mini-white' gene to produce coloured eyes. For each strain, I designed experiments using females and males that had been given the opportunity to mate with each other for 48 hours before being separated into single-sex conditions, to mimic the conditions commonly used in experiments designed to investigate ageing [21].

I first assessed the lifespan of *CS*, *Dah*, and $w^{1118}$ mated female and male flies at 25˚C on a 12h:12h light:dark cycle (details in **Materials and Methods**). Within each strain, I observed that males consistently showed shorter median lifespan than females, albeit with a significant difference in the proportional lifespan difference between females and males among strains

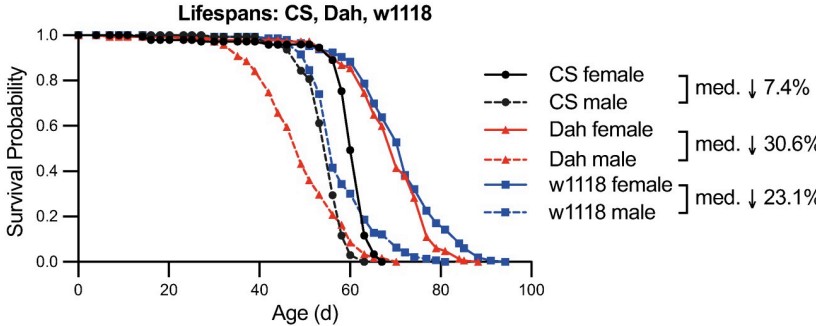

**Fig 1. Lifespan profiles of *CS*, *Dah*, and *w^1118* female and male flies.** Survival curves for each sex and strain show shorter median lifespan for male flies compared with female flies within each strain, with percent decreases in median lifespan listed. Cox Proportional Hazards analysis for the interaction between sex and strain finds p = 0.00036. For each pairwise comparison between sexes within a strain, log-rank tests find p<10^{-20}. For each pairwise comparison between strains within a sex, log-rank tests find p<10^{-6}, except for *Dah* female vs *w^1118* female (p = 0.016) and *Dah* male versus *CS* male (p = 0.00049). n>125 deaths counted per condition.

(Fig 1, Cox Proportional Hazards p = 0.00036 for interaction between sex and strain). For instance, *CS* males showed a comparatively modest 7.4% decrease in median lifespan compared to *CS* females, whereas *Dah* and *w^1118* males showed more substantial 30.6% and 23.1% decreases in median lifespan respectively. Interestingly, the more isogenic *CS* strain also showed more uniform ages at death with 'steeper' survival curves, particularly compared with the outbred *Dah* strain that has been maintained in large populations since its original collection from the wild. Taken together, these data indicate that not only absolute lifespan but also the degree to which sex modulates lifespan varies significantly among *Drosophila* background strains.

## Age-related changes in activity levels depend on strain and sex

I next assessed how behavioural activity levels during the light (day) and dark (night) cycles change with age among strains and sexes. I collected mated female and male flies of each strain at 2, 4, 6, or 8 weeks of age and used the Trikinetics *Drosophila* Activity Monitor (DAM5H) system to collect activity counts per minute in 12h:12h light:dark (LD) conditions. Plotting the mean activity counts in each 30-minute window of the 24-hour LD cycle (Fig 2A–2F) showed strikingly divergent effects of age on behaviour among strains and sexes. Surprisingly, several populations showed significantly increased activity levels in 6-week and/or 8-week-old flies compared to 2-week-old flies (Fig 2G–2L). Specifically, aged *CS* female and male flies showed significant increases in activity during both day and night cycles (Fig 2G and 2J), whereas *w^1118* females showed significant increases in day, but not night, activity (Fig 2I and 2L). In contrast to previous studies showing moderate reductions in activity for aged *CS* flies [6], the only populations showing any age-related reduction in activity were *Dah* and *w^1118* males, which showed reduced night, but not day, activity at 8 weeks (Fig 2K and 2L). To determine whether these results would be reproducible in independent experiments, I ran replicate experiments for female and male flies of each strain at 2 and 6 weeks of age. Here, I again observed increased activity levels in many older populations, with significantly increased day activity for each female population as well as *w^1118* males, and increased night activity for *CS* males and females as well as *w^1118* males (S2 Fig). While potentially unexpected, these data indicate that age-related reduction in activity level is not a universally shared phenotype among strains and sexes.

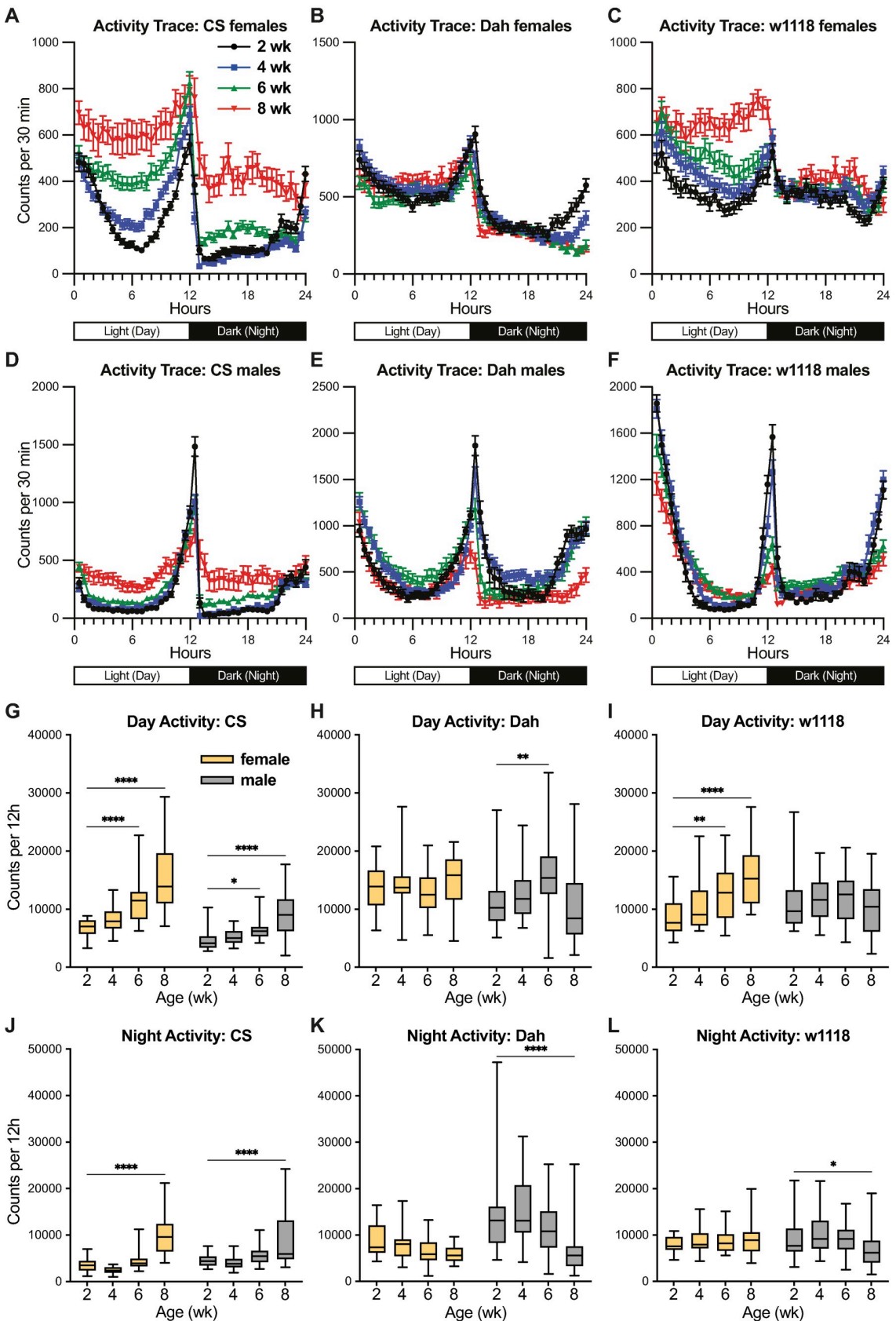

**Fig 2. Activity profiles of *CS*, *Dah*, and *w^1118* female and male flies.** (A-F) Activity traces show the population mean +/- SEM of activity counts for each 30-minute bin across day and night cycles. (G-L) Box-and-whisker plots (minimum, 25%, median, 75%, maximum) show the summed activity counts during the day and night cycles (mean across recording days) for flies of each strain, sex, and age. n = 17–32 flies per condition; * p<0.05, ** p<0.01, *** p <0.001, **** p<0.0001 by Dunnett's multiple comparisons test versus 2-week-old group. 2-way ANOVA results are reported in S1 Fig.

## Age-related changes in anticipation indices are largely shared among strains and between sexes

In addition to overall activity levels, day/night patterns of activity have previously been reported to change with age [22,23]. *Drosophila melanogaster* individuals generally show a crepuscular pattern of activity, with higher activity levels near dusk and dawn. This pattern includes 'anticipation' that can be measured by an increase in activity in the hours immediately preceding the lights-on or lights-off transitions. Consistent with previous reports [22,23], I observed that young populations of flies show robust patterns of anticipation, with activity peaks preceding the lights-on morning and lights-off evening transitions (Fig 3A); the strength of these anticipation patterns can be quantified using published anticipation index metrics [24].

When quantifying the morning and evening anticipation indices for each population of flies among my experiments, I observed that every sex and strain showed significant decreases in the morning anticipation index by 8 weeks of age, with *CS* and *Dah* females and males showing significant decreases starting even at 4 weeks of age compared to 2 weeks (Fig 3B–3D). While I did not see as universal a pattern in age-related changes for the evening anticipation index in these experiments, I observed significant reductions in this measure by 6 weeks of age for *CS* females and males, *Dah* males, and *w^1118* males (Fig 3E–3G). Replicate experiments for 6-week versus 2-week old flies (S3 Fig) showed similar results, but with stronger age-related changes in evening anticipation (significantly decreased by 6 weeks for all populations except *CS* females) than for morning anticipation. These results are particularly striking when considered in light of the strong divergence in age-related changes in overall activity levels among sexes and strains (Figs 2 and S2). Despite marked differences in whether populations became more or less active with age, almost all populations showed robust age-related declines in their ability to anticipate the morning lights-on and/or evening light-off transition as measured by their anticipation indices.

## Age-related changes in sleep quantity depend on strain and sex

In addition to activity patterns, sleep quantity and quality have been useful metrics of behavioural ageing in *Drosophila* [6,25,26]. To assess sleep quantity in my experimental populations, I used a widely accepted proxy measure of sleep in *Drosophila* activity assays, namely five or more consecutive minutes with zero activity counts [27,28]. When plotting the average amount of sleep in each 30-minute window of the 24-hour LD cycle (Fig 4A–4F), I observed very striking differences among populations. For instance, males from all strains, as well as *CS* females, showed sleep patterns consistent with a large body of published work: these flies tended to show high wakefulness in the morning, move to a period of increased sleep in the middle of the day, then show high wakefulness again in the evening before a more consolidated period of sleep at night (Fig 4A and 4D–4F). In contrast, *Dah* and *w^1118* females showed very little sleep compared to other populations, with *w^1118* females showing little clear pattern of sleep between day and night cycles (Fig 4B and 4C). When quantifying age-related changes in sleep amount (Fig 4G–4L), I also observed striking differences among strains and sexes. Potentially consistent with the age-related hyperactivity I observed in *CS* females and males

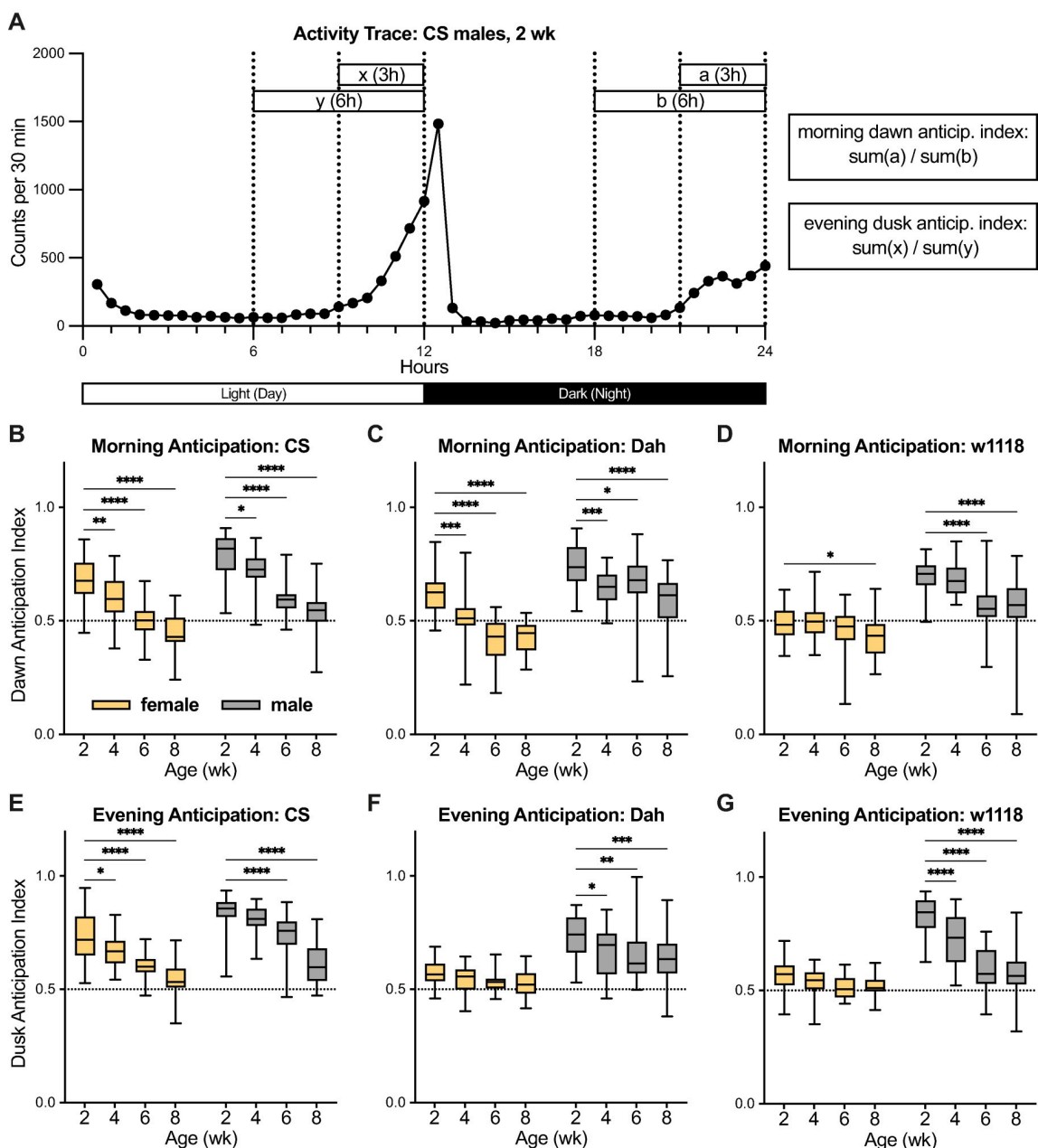

**Fig 3. Morning and evening anticipation indices in *CS*, *Dah*, and *w^1118* female and male flies.** (A) Example activity trace (reproduced from Fig 2D) shows the method by which morning (dawn) and evening (dusk) anticipation indices are calculated. (B-G) Box-and-whisker plots (minimum, 25%, median, 75%, maximum) show the morning or evening anticipation indices (mean across recording days) for flies of each strain, sex, and age. n = 17–32 flies per condition; * p<0.05, ** p<0.01, *** p <0.001, **** p<0.0001 by Dunnett's multiple comparisons test versus 2-week-old group. 2-way ANOVA results are reported in S1 Fig.

(Fig 2G and 2J), I observed significant age-related decreases in sleep amount in *CS* females and males by 6 weeks of age (Fig 4G and 4J). At the same time, this correlation between age-related changes in activity levels and age-related sleep amount was not shared among all populations: for instance, aged *w^1118* males showed no clear change in daytime activity levels with increasing age (Fig 2I), but this same population showed strongly significant age-related decreases in daytime sleep with increasing age (Fig 4I). In replicate experiments, I observed similar patterns

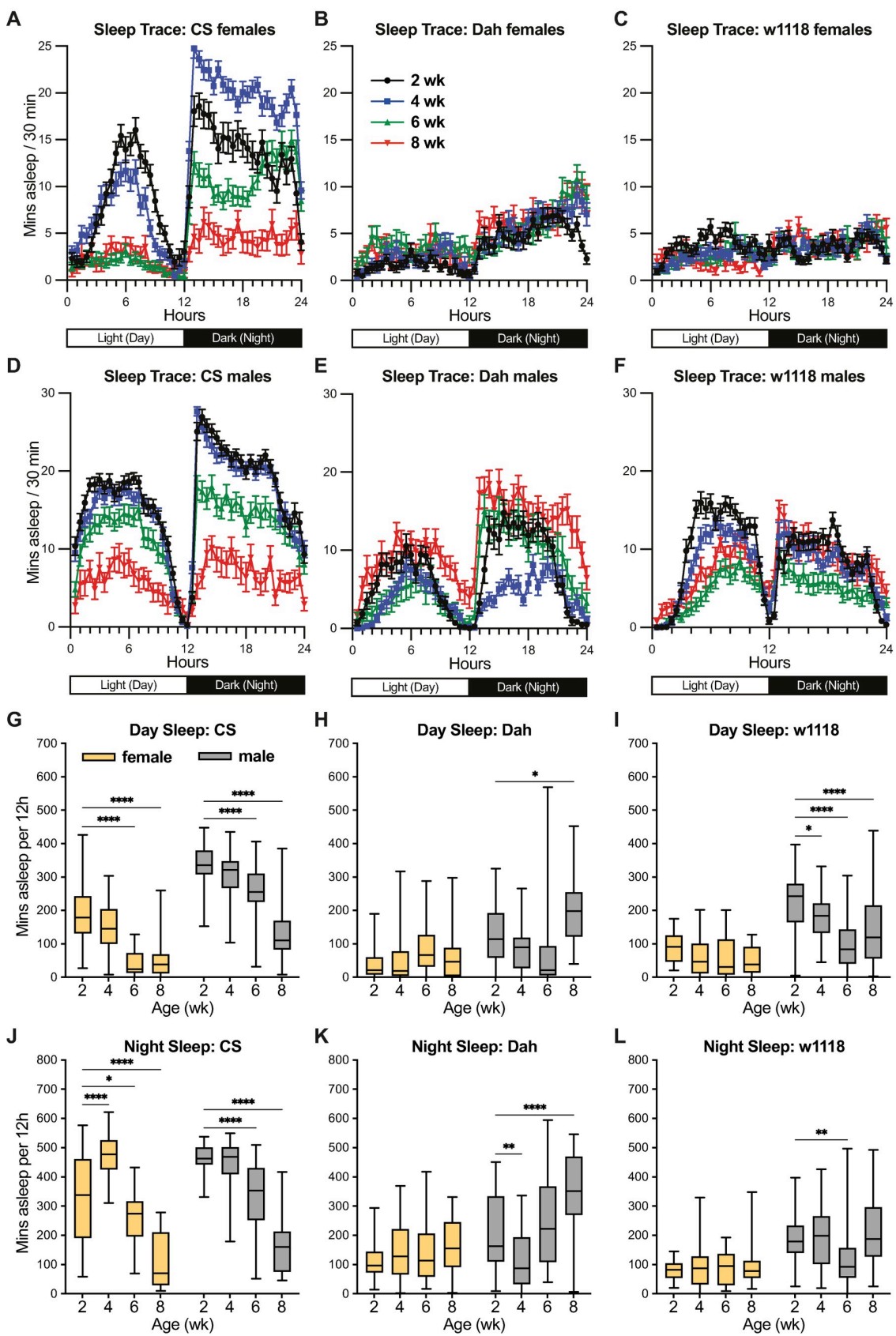

**Fig 4. Sleep profiles of *CS*, *Dah*, and *$w^{1118}$* female and male flies.** (A-F) Sleep traces show the population mean +/- SEM of sleep minutes (defined as 5 or more consecutive minutes of 0 activity counts) for each 30-minute bin across day and night cycles. (G-L) Box-and-whisker plots (minimum, 25%, median, 75%, maximum) show the summed sleep minutes during the day and night cycles (mean across recording days) for flies of each strain, sex, and age. n = 17–32 flies per condition; * p<0.05, ** p<0.01, **** p<0.0001 by Dunnett's multiple comparisons test versus 2-week-old group. 2-way ANOVA results are reported in S1 Fig.

among strains and sexes: *Dah* and *$w^{1118}$* females showed markedly less sleep than other populations; *CS* females and males showed reduced day and night sleep with age; and age-related sleep changes for *Dah* and *$w^{1118}$* flies differed between sexes (S4 Fig). These findings indicate that, while sleep quantity offers a distinct behavioural measure from activity levels, neither measure shows consistent changes with age across the strains and sexes in these experiments.

## Age-related changes in sleep quality depend on strain and sex

Sleep quality–namely the way sleep is consolidated into longer bouts (individual sleep episodes) or fragmented into shorter bouts–has been one of the most widely studied measures of age-related behavioural change in *Drosophila*, with studies generally reporting that older flies have higher numbers of sleep bouts with shorter sleep bout duration (indicating more fragmented sleep), particularly during the night [6,25,26]. To assess sleep quality among the experimental populations here, I first assessed the mean number of day and night sleep bouts among ages, strains, and sexes (Fig 5). Here, I observed a decrease in the number of day sleep bouts with age for most strains and sexes, but no consistent pattern for night sleep bouts among strains and sexes: for instance, *CS* females and males displayed decreasing numbers of night sleep bouts at the oldest ages (5D), whereas *Dah* males displayed increased numbers of night sleep bouts at the oldest ages (5E). In replicate experiments performed at 2 weeks and 6 weeks of age, I observed similar overall patterns: most strains and sexes exhibited a reduction in the number of day sleep bouts at older ages, but patterns for night sleep bouts were less clear (S5 Fig).

To consider another measure of sleep quality, I next assessed the mean sleep bout length among ages, strains, and sexes (Fig 6). Here, I observed some patterns consistent with previously published studies: for instance, 6-week and 8-week-old *CS* females and males all showed shorter sleep bout length during both the day and night as compared to 2-week-old flies (Fig 6A and 6D). However, I did not observe a similar pattern in either *Dah* or *$w^{1118}$* flies, suggesting a high level of divergence in how sleep quality changes with age among *Drosophila* background strains. In replicate experiments, I observed some similar patterns of reduced day sleep bout length in older *CS*, *Dah*, and *$w^{1118}$* males, and reduced night sleep bout length in older *CS* males; however, all remaining groups including all females showed increased or unchanged sleep bout length with age (S6 Fig). Taken together with my data on sleep quantity (Fig 4), these analyses do not suggest that sleep patterns change with age in a consistent way among strains and sexes; instead, they suggest that the measures of sleep and activity most relevant for ageing may depend heavily on factors such as background strain and sex.

## Discussion

Increasingly, research on ageing using model organisms has considered both lifespan (as measured by survival) and healthspan (as measured by changes in behaviour and other phenotypes with age) [5]. *Drosophila* has been a productive model system for investigating sex differences in both lifespan and healthspan, as well as the sex-specific plasticity of these phenotypes with longevity-promoting interventions [29,30]. With respect to lifespan, my current results are consistent with many of these past studies: in each strain investigated, I observed significantly

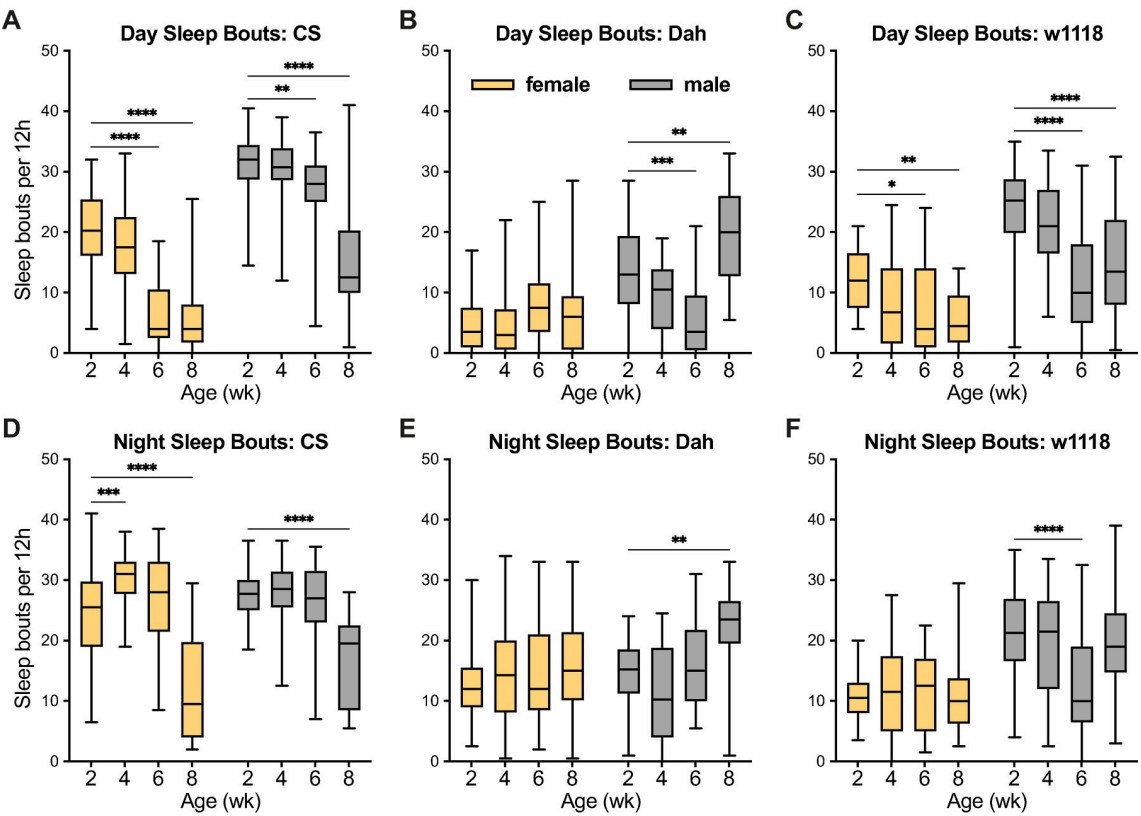

**Fig 5. Sleep bout numbers in *CS*, *Dah*, and *w1118* female and male flies.** (A-F) Box-and-whisker plots (minimum, 25%, median, 75%, maximum) show the number of independent sleep bouts (defined as 5 or more consecutive minutes of 0 activity counts) during the day and night cycles (mean across recording days) for flies of each strain, sex, and age. n = 17–32 flies per condition; * $p<0.05$, ** $p<0.01$, *** $p<0.001$, **** $p<0.0001$ by Dunnett's multiple comparisons test versus 2-week-old group. 2-way ANOVA results are reported in S1 Fig.

greater longevity in female compared to male flies (Fig 1). However, the extent to which females outlived males differed greatly among strains, suggesting that studies investigating sex differences in ageing may come to slightly different conclusions depending on the background strain used. Importantly, my lifespan results differ from one of the foundational studies on age-related sleep and activity changes in *Drosophila*, where the authors observed longer lifespan in male than in female *CS* flies [6]. This suggests that additional factors, for example diet and specific rearing conditions, likely contribute to sex differences in lifespan. Even when diet and rearing conditions are controlled, genetic background can play a major role in sex differences in lifespan: for instance, among strains in the *Drosophila* Genetic Reference Panel, previous studies have observed significant genetic variation for sexual dimorphism of lifespan, with some strains showing longer lived males and some showing longer lived females [31]. Taken together with my current results, these studies highlight that sexual dimorphic effects on ageing need to be considered in a context- and strain-specific manner.

With respect to healthspan, my data only partially overlap with previous studies, and they highlight some important differences among strains and sexes. Firstly, my data on age-dependent sleep fragmentation (as measured by decreased sleep bout length) parallel only a subset of results from previous studies. For instance, one of the first studies to thoroughly investigate age-related sleep change in flies found that *CS* virgin female and male flies show a transient increase in sleep bout length in early life, followed by a decline in sleep bout length in late life

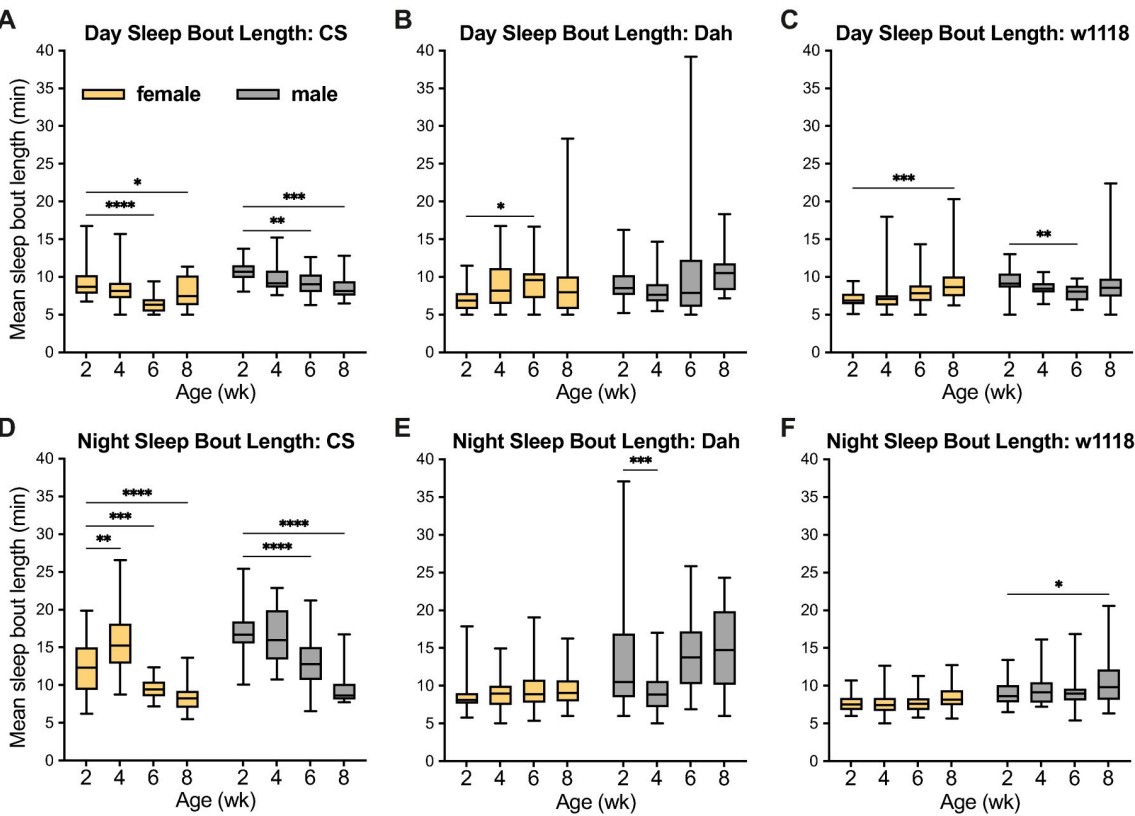

**Fig 6. Sleep bout length in *CS*, *Dah*, and *w*<sup>1118</sup> female and male flies.** (A-F) Box-and-whisker plots (minimum, 25%, median, 75%, maximum) show the length of sleep bouts (defined as 5 or more consecutive minutes of 0 activity counts) during the day and night cycles (mean across recording days) for flies of each strain, sex, and age. n = 17–32 flies per condition; * $p < 0.05$, ** $p < 0.01$, *** $p < 0.001$, **** $p < 0.0001$ by Dunnett's multiple comparisons test versus 2-week-old group. 2-way ANOVA results are reported in S1 Fig.

[6]. My current results for mated female and male *CS* flies parallel this previous study, but my results for *w*<sup>1118</sup> and *Dah* flies of both sexes are markedly different (Fig 6). In addition, I observe strong age-related increases in activity counts for *CS* flies (Fig 2) which were not observed in the previous study [6]. Other more recent studies have examined both *CS* and *w*<sup>1118</sup> flies and found that both strains exhibited age-dependent declines in sleep bout length [25,26], as has another study examining virgin female flies of the *w*<sup>Dah</sup> strain (in which the *w*<sup>1118</sup> allele has been backcrossed into an otherwise *Dah* genetic background) [18]. However, yet other studies have failed to observe age-dependent shortening of sleep bout length in male flies derived from a *CS* background [32]. Taken together, these results highlight sex- and strain-dependent discrepancies among previous literature and my own results on age-dependent increases in sleep fragmentation.

At the same time, my data highlight other behavioural phenotypes, such as anticipation indices, that more closely align with previous findings and show robust age-dependent changes across sexes and strains. Previous work has shown age-dependent reductions in both morning and evening anticipation indices for *w*<sup>1118</sup>-*iso*<sup>31</sup> female and male flies [22,23], effects that are largely reproduced by my own results for these measures in almost all strains and sexes examined here (Figs 3 and S3). As noted in this previous work, morning anticipation behaviour is controlled by a very specific subsets of neurons that express the neuropeptide Pdf [33,34]. Moreover, Pdf levels decrease with age in *CS* males [35], and RNAi-based knockdown of *Pdf* in ventral lateral neurons reduces morning anticipation in young male flies [36]. Taken

together, these findings present an attractive hypothesis in which age-dependent loss of Pdf in a specific class of neurons directly leads to age-dependent decline in anticipation. When coupled with my current findings of declining morning and/or evening anticipation in females and males from three different strains, these data suggest that reduced Pdf expression and a resulting decline in anticipation have the potential to be robust phenotypes of ageing that might occur broadly across sexes and strains.

What factors might contribute to the discrepancies between the current results and previous studies for some but not all phenotypes? For instance, some of my sleep data here, particularly for *Dah* and $w^{1118}$ females (Fig 4B and 4C) show much lower overall sleep levels and reduced day/night distinctions in sleep patterns than other strains/sexes in either the current study or previous studies that used similar strains, for example mated $w^{1118}$ females [25]. These differences appear to be intrinsic to the strains used in the present study, rather than to experiment-specific effects, as the same patterns appear for *Dah* and $w^{1118}$ females in replicate experiments (S4B and S4C Fig). Some of these differences may be due to diet, rearing conditions, and genetic/epigenetic divergence among 'the same' strains that have been kept in different labs over the course of tens to hundreds of generations (see Methods for details on the stocks used here); this suggests that caution should be used when comparing results among studies, even when they use strains that originated from the same source many years ago. Another major difference in the current study may lie in the nature of the activity monitors used: many previous studies have used the Trikinetics DAM2 system in which a single infrared beam passes through each single-fly tube, whereas the data in the current study are derived from the newer generation Trikinetics DAM5H system in which 15 infrared beams pass through each tube. In theory, the DAM5H system should allow for finer-scale movement detection even when flies move within a small region of the tube–meaning that these data will likely report higher activity and lower sleep levels than previous studies. Indeed, other studies comparing video tracking approaches to DAM2-based assays have shown that DAM2-based analyses overestimate sleep duration and sleep bout length because they miss many brief movements better captured by video tracking [37,38]; further developments using video tracking approaches [39,40] have even allowed more recent studies to identify with great precision that some individual wild-type flies are effectively sleepless [41]. However, it should be noted that my own data in DAM5H systems do show similar age-related changes in activity and sleep for *CS* flies as compared to previous studies using DAM2 systems [6], and other studies comparing single-beam, multi-beam, and video tracking systems have found that all three approaches identify similar patterns of age-dependent sleep fragmentation within $iso^{31}$ flies [37]. This all being said, phenotypes in which data are normalised to each fly's activity level (such as the anticipation indices that employ a ratio rather than absolute numbers) may be even more reproducible among recording systems and potentially among sexes and strains– perhaps explaining why these measures show more similar age-dependent trajectories between previous studies and the current work. As one potential downside of the DAM5H system, some users have reported spurious activity counts from oversensitive detectors, even in empty tubes; however, in experiments where I included a set of empty tubes in parallel with the fly studies shown here, I was not able to detect any spurious counts in my experimental setup (n = 120 beams, 8 tubes; 0 counts for all beams across >90 hours of recording).

Additional factors may also contribute to seemingly discordant findings between this work and previous studies. *Drosophila* diet, including the specific type of each ingredient, can have major effects on age-related phenotypes [42]; even the specific lot number of yeast from the same company can alter *Drosophila* phenotypes, as can the type of preservative used [43]. The *Drosophila* diet used in the current study is very similar to some previous studies on age-related sleep changes [18] but distinct from others [6,22,23,25,26]; diet may therefore be an

additional factor explaining differences among studies. Sex and mating status can also have profound effects on age-related phenotypes in *Drosophila*: for instance, females display higher levels of gut hyperplasia and permeability with age [29,44] in a manner dependent on both mating status [45] and background strain [29]. My current results may therefore compare most easily with other studies using mated female and male flies rather than those that have used virgin flies. Importantly, my results also find significant age-by-sex interactions for all measures of behaviour in at least one of the strains tested here (S1 Fig). This is not surprising, as sexual dimorphism in circadian activity and sleep patterns are well established in the literature: for example, across multiple *Drosophila* strains, males exhibit markedly greater mid-day sleep than do females [28], in a behaviour pattern termed the 'siesta'. The cellular, physiological, and anatomical underpinnings of these behaviours are also sexually dimorphic: within the example of siesta behaviour, a subset of sleep-promoting dorsal clock neurons (DN1) show increased daytime activity in males compared to females [46], and signals such as male sex-peptide-derived effects in mated females [47] and sex-determination signalling pathways in mushroom body neurons and/or fat body cells [48] can also contribute to sexual dimorphism in the siesta behaviour. Given both my current results and well-established examples of sexual dimorphism in *Drosophila* activity and sleep behaviour, it is highly likely that behavioural studies will uncover different or even opposing effects of anti-ageing interventions in females versus males. Finally, while this work has used three different strains in an attempt to find robust phenotypes across *Drosophila*, this is still a relatively limited number of strains and will not reflect the full range of age-related behavioural changes among the large number of common *Drosophila* lab strains.

Given the strong divergence of age-related behavioural change among strains and sexes, what conclusions can be drawn from existing and future studies using these measures as correlates of healthspan? Firstly, and as a note of moderation, the current data should not necessarily call into question previous studies using *Drosophila* to examine age-related sleep or activity changes. Well-designed studies have offered important molecular mechanistic insight into genetic or pharmacological interventions that extend lifespan and delay behavioural features of ageing like sleep fragmentation within a strain and/or sex [18]. Future investigation into whether these mechanisms extend to other strains and sexes will therefore be important follow-up research to understand how universal these mechanisms are. Secondly, the current results suggest that some measures of behaviour–for example morning or evening anticipation–could potentially offer more reproducible measures of age-related behavioural change among strains and between sexes. Finally, these results reinforce the importance of backcrossing and other strategies to control for background strain in studies examining mutant or transgenic animals, as the background strain of each mutant or transgenic line may differ greatly in behaviour and therefore lead to spurious results. Control of genetic background has long been recognised as an important benchmark for research on ageing [49] and other physiological quantitative traits [50]; the current data therefore underscore the importance of proper controls in genetic studies. Importantly, background strain effects are likely to affect studies in mammals as well, as age-related behavioural changes can also be strain-dependent in mice [51]. Ultimately, these results suggest a note of caution should be used when drawing conclusions from even carefully controlled behavioural studies on single sexes or background strains in any model organism.

## Materials and methods

### Fly stocks and husbandry

*Drosophila* stocks were maintained and experiments conducted at 25˚C on a 12h:12h light: dark cycle on sucrose-yeast-agar (SYA) food containing 10% (w/v) brewer's yeast (MP Bio, lot

number U1122284494-1), 5% (w/v) sucrose, 1.5% (w/v) agar, 0.3% (w/v) Nipagin, and 0.3% (v/v) propionic acid.

The *Canton-S* (*CS*) wild-type stock was originally isogenised in 1943 [52] and has been widely used in neurogenetics research since being chosen by Seymour Benzer for its homogeneous phototactic behaviour [53]. The *CS* stock used here was a gift from Colin McClure (Queen's University Belfast); this stock was maintained in the McClure lab for more than 4 years after being obtained from the lab of Tony Southall (Imperial College London). The wild-caught *Dahomey (Dah)* wild-type stock was collected in 1970 in Dahomey (now Benin) and has since been maintained in large populations with overlapping generations; the *Dah* stock used here was obtained from the lab of Linda Partridge (University College London), where it has been maintained for more than 20 years. The $w^{1118}$ stock used here was also obtained from the Partridge lab, where it was initially obtained from the Bloomington Stock Center [54] and has been kept in the Partridge lab for more than 15 years. All stocks were maintained under identical conditions (multiple large population bottles, SYA food, 25˚C) for at least three generations in the lab of Nathan Woodling, University of Glasgow, before being used in the present studies.

## Survival analysis

Lifespan assays were carried out as described in detail in [21]. From the eggs collected from each set of parental flies, the progeny that emerged as adults within a 24-hour window were collected and allowed to mate for 48 hours, after which they were separated into single-sex vials containing SYA fly food at a density of 15 individuals per vial. Flies were transferred to fresh vials three times per week, with deaths and censors scored during each transfer. Microsoft Excel (template available at http://piperlab.org/resources/) was used to calculate survival proportions after each transfer.

## Drosophila activity monitor experiments

For the activity monitor experiments shown in Figs 2–6, flies were generated and maintained as for survival analysis above, with successive groups of mated female and male flies generated at 2-week intervals. When the oldest cohort of flies for a strain reached 8 weeks of age, all four cohorts (2-week, 4-week, 6-week, and 8-week) were briefly anaesthetised and transferred to single-fly tubes (Trikinetics, 65mm glass tubes) containing a small amount of SYA fly food at one end and a small cotton plug at the other end. These tubes were then inserted into DAM5H Drosophila Activity Monitors (DAMs, Trikinetics) with the food adjacent to beam number '1'. DAMs were then placed in an incubator at 25˚C on a 12h:12h light:dark cycle for at least 72 hours (sufficient time to gather data before any fertilised eggs produced larvae that could confound activity counts). Data were collected using the DAMSystem3 software (Trikinetics) with activity counts collected once each minute. All flies were allowed at least 24 hours to acclimatise to the DAMs; data were then extracted for 48 hours starting from the next lights-on event. During these 48 hours of data recording, the exact ages of each cohort were 13–14 days, 27–28 days, 41–42 days, and 55–56 days.

For the activity monitor experiments shown in S2–S6 Figs, flies were generated and maintained as above, with successive groups of mated female and male flies generated at a 4-week interval. When the older cohort of flies for a strain reached 6 weeks of age, both cohorts (2-week and 6-week) were briefly anaesthetised and transferred to DAM5H monitors as above. All flies were allowed at least 24 hours to acclimatise to the DAMs; data were then extracted for 24 hours starting from the next lights-on event. During these 24 hours of data recording, the exact ages of each cohort were 13 days and 41 days.

Data on activity counts were analysed using R (scripts available at Mendley Data, https://doi.org/10.17632/8633hm46p5.1) in RStudio Version 2023.03.1+446. These scripts were designed to reproduce previous Excel-based analysis pipelines described in [55]. Briefly, the first script translated Trikinetics monitor files containing per-beam counts to files containing total activity counts summed over all beams. The second script calculated per-fly measures of activity counts (total, day, and night); anticipation indices (morning and evening); sleep amounts (total, day, and night); sleep bout number (total, day, and night); and sleep bout length (total, day, and night). For all experiments, sleep was defined as 5 or more consecutive minutes of inactivity, as widely defined in the literature [27,28]. Morning and evening anticipation indices were calculated as shown in Fig 3A, as previously defined [24]. This script also excluded any flies that died during the recording; for these experiments, deaths were defined as flies with fewer than 3 activity counts during the final 180 minutes of analysed recording. The third script compiled data from across different monitor files to create a single table for each measure across experimental conditions. Analysed data were then exported to GraphPad Prism 10.0 for graphical display and statistical analysis.

## Statistical analysis

The graphical displays (generated in GraphPad Prism 10.0) and statistical tests used for each readout are described in the associated figure caption. Log-rank tests were performed in Microsoft Excel (template available at http://piperlab.org/resources/); Cox Proportional Hazards tests were performed in R using the 'survival' package; and 2-way ANOVA with Dunnett's or Bonferroni multiple comparisons tests were performed in GraphPad Prism 10.0. For all statistical tests, $p < 0.05$ was considered significant.

## Supporting information

**S1 Fig. 2-way ANOVA results for all figures.**
(PDF)

**S2 Fig. Activity profiles of 2- and 6-week-old *CS*, *Dah*, and *w^1118* female and male flies, replicate experiments.** (A-F) Activity traces show the population mean +/- SEM of activity counts for each 30-minute bin across day and night cycles. (G-L) Box-and-whisker plots (minimum, 25%, median, 75%, maximum) show the summed activity counts during the day and night cycles for flies of each strain, sex, and age. n = 41–64 flies per condition; * $p < 0.05$, ** $p < 0.01$, **** $p < 0.0001$ by Bonferroni multiple comparisons test.
(PDF)

**S3 Fig. Morning and evening anticipation indices in 2- and 6-week-old *CS*, *Dah*, and *w^1118* female and male flies, replicate experiments.** (A-F) Box-and-whisker plots (minimum, 25%, median, 75%, maximum) show the morning or evening anticipation indices for flies of each strain, sex, and age. n = 41–64 flies per condition; ** $p < 0.01$, **** $p < 0.0001$ by Bonferroni multiple comparisons test.
(PDF)

**S4 Fig. Sleep profiles of 2- and 6-week-old *CS*, *Dah*, and *w^1118* female and male flies, replicate experiments.** (A-F) Sleep traces show the population mean +/- SEM of sleep minutes (defined as 5 or more consecutive minutes of 0 activity counts) for each 30-minute bin across day and night cycles. (G-L) Box-and-whisker plots (minimum, 25%, median, 75%, maximum) show the summed sleep minutes during the day and night cycles for flies of each strain, sex, and age. n = 41–64 flies per condition; * $p < 0.05$, ** $p < 0.01$, *** $p < 0.001$, **** $p < 0.0001$ by

Bonferroni multiple comparisons test.
(PDF)

**S5 Fig. Sleep bout numbers in 2- and 6-week-old *CS, Dah*, and *w^1118* female and male flies, replicate experiments.** (A-F) Box-and-whisker plots (minimum, 25%, median, 75%, maximum) show the number of independent sleep bouts (defined as 5 or more consecutive minutes of 0 activity counts) during the day and night cycles for flies of each strain, sex, and age. n = 41–64 flies per condition; ** p<0.01, *** p <0.001, **** p<0.0001 by Bonferroni multiple comparisons test.
(PDF)

**S6 Fig. Sleep bout length in 2- and 6-week-old *CS, Dah*, and *w^1118* female and male flies, replicate experiments.** (A-F) Box-and-whisker plots (minimum, 25%, median, 75%, maximum) show the length of sleep bouts (defined as 5 or more consecutive minutes of 0 activity counts) during the day and night cycles for flies of each strain, sex, and age. n = 41–64 flies per condition; * p<0.05, ** p<0.01, *** p <0.001, **** p<0.0001 by Bonferroni multiple comparisons test.
(PDF)

## Acknowledgments

I am grateful for the diligent and professional work from the staff of the University of Glasgow Academic Service Unit in their preparation of the *Drosophila* media needed to run these experiments. I also thank the labs of Adam Dobson and Alberto Sanz Montero for equipment sharing and scientific advice throughout this project.

## Author Contributions

**Conceptualization:** Nathan Woodling.

**Formal analysis:** Nathan Woodling.

**Funding acquisition:** Nathan Woodling.

**Investigation:** Nathan Woodling.

**Methodology:** Nathan Woodling.

**Project administration:** Nathan Woodling.

**Visualization:** Nathan Woodling.

**Writing – original draft:** Nathan Woodling.

**Writing – review & editing:** Nathan Woodling.

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
