## [Decision Letter · Decision Letter 0]

19 Mar 2024

PONE-D-24-06034Sex- and strain-dependent effects of ageing on sleep and activity patterns in DrosophilaPLOS ONE

Dear Dr. Woodling,

Thank you for submitting your manuscript to PLOS ONE. After careful consideration, we feel that it has merit but does not fully meet PLOS ONE’s publication criteria as it currently stands. Therefore, we invite you to submit a revised version of the manuscript that addresses the points raised during the review process.

Although the reception of your manuscript was generally favorable, reviewer 1 raises a significant issue regarding rigor/replicability, which along with the remaining comments and discussion suggestions should be thoroughly addressed. Looking forward to the revised manuscript

We look forward to receiving your revised manuscript.

Kind regards,

Efthimios M. C. Skoulakis, PhD

Academic Editor

PLOS ONE

Journal Requirements:

Reviewers' comments:

Reviewer's Responses to Questions

**Comments to the Author**

1. Is the manuscript technically sound, and do the data support the conclusions?

Reviewer #1: Partly

Reviewer #2: Yes

2. Has the statistical analysis been performed appropriately and rigorously? 

Reviewer #1: Yes

Reviewer #2: Yes

3. Have the authors made all data underlying the findings in their manuscript fully available?

Reviewer #1: Yes

Reviewer #2: Yes

4. Is the manuscript presented in an intelligible fashion and written in standard English?

Reviewer #1: Yes

Reviewer #2: Yes

5. Review Comments to the Author

Reviewer #1: In this manuscript the author compares longevity, locomotor activity, and sleep between three strains of Drosophila melanogaster and between males and females for each strain. The results suggest that, particularly regarding age associated changes in sleep, that strains vary significantly in the extent to which sleep changes over time and that major differences are observed between the sexes within these strains, particularly with regard to changes in sleep as flies age. The study contrasts these strain and sex specific differences in sleep with age associated changes in anticipatory locomotor activity, which appears to be characterized by less strain-to-strain variability. The study offers a useful cautionary reminder that age-related changes in sleep will be highly dependent on the strain and sex of the flies studied. However, there are a few concerns that might be addressed by the author to improve the study.

The major concern here is the apparent absence of replication. For all behavioral experiments a sample size of “17-32” is reported for each strain and sex. This suggests that a single monitor was run for each strain/sex, leaving open the question of how replicable the strain and sex differences described were in the hands of the author. This significantly limits the impact of the study. This is particularly worrying given that, in several instances, the author describes extremely surprising results that do not fit previously published work. The data shown in figures 4B and C are striking examples, in which female Dah and w1118 flies appear to be nearly insomniac. This is in stark contrast to a significant amount of previous work with the w1118 strain. One wonders if something strange happened during the single run with these flies.

Should these strain/sex differences survive replication, it is important to note that such strains are evolving independently in the various labs in which they are being reared. The differences described here could be unique to the author’s lab. This should be noted when discussing differences from previously published results. Related to this, it would be good if the author included information about how long ago the three strains were established in the labs that provided the flies used in this study.

The author should more fully describe the differences between this study and previously published sleep data (there are a great number of studies reporting sleep in CS and w1118 strains and much of it looks very different than what is reported here). This is done to some extent, but the author should spell out what is different about the general profiles and amounts of sleep observed.

The author should more fully describe the existing literature on sexual dimorphism in Drosophila sleep and circadian timekeeping and place the results described in this context.

Other Suggestions:

Line 84: It is not clear what “standard physiological conditions” are.

SEM or some other indication of error should be indicated on the averaged activity and sleep plots shown in Figures 2A-F and 4A-F.

The author should reconsider the logic of the grammar used in lines 202 and 220 which appear to suggest that previously published work came AFTER this manuscript.

The discussion at the top of page 9 is useful, but the author should acknowledge previous work establishing that single beam activity monitoring overestimates fly sleep (when compared to video tracking or multi-beam monitoring).

Line 325: the author states that “standard fly food” was used. There is no such thing. The author acknowledges elsewhere that diet has major effects on aging (and sleep is quite sensitive to it, too). The author should therefore define the media used (both in vials and in capillary tubes).

Reviewer #2: The author completed a thorough analysis of age-related changes to sleep in multiple commonly-used Drosophila strains of flies, and in both sexes. The goal of this analysis was to provide a unifying view of potential reasons for discrepancies between studies using different sexes and strains regarding age-related changes to sleep. The author found that sex differences in sleep and age-related changes to sleep were not the same across strains. The author also found that age-related changes to sleep were not the same across strains. This suggests that both sex and strain matter when studying sleep and age-related changes to sleep, and that in future conclusions must be interpreted with caution in cases where multiple strains or both sexes have not been used. This study will prompt increased rigour in the field and represents a very important contribution. I am also glad the authors have found that females have a longer lifespan, this is something that is common yet difficult to publish considering the one study that reported a longer male lifespan.

6. PLOS authors have the option to publish the peer review history of their article (what does this mean?). If published, this will include your full peer review and any attached files.

Reviewer #1: No

Reviewer #2: No

---

## [Author Response · Author response to Decision Letter 0]

3 Jul 2024

1 July 2024

Dear Editor,

Thank you for the opportunity to submit here a revised version of manuscript PONE-D-24-06034, “Sex- and strain-dependent effects of ageing on sleep and activity patterns in Drosophila,” to be considered for publication as a research article in PLOS ONE. I have addressed the editorial and reviewer comments as outlined below (my responses preceded by dashes), with changes in the ‘with changes’ version of the manuscript highlighted in yellow.

---

Editorial Comments

Although the reception of your manuscript was generally favorable, reviewer 1 raises a significant issue regarding rigor/replicability, which along with the remaining comments and discussion suggestions should be thoroughly addressed. Looking forward to the revised manuscript

- I agree that the point about replicability was an important one, particularly given that my results differ significantly from some previously published studies. As outlined in the responses to Reviewer 1, I have now provided results from larger-scale replicate experiments in the Supplementary Data that I hope address their concerns. I have also addressed the reviewer’s other concerns with additional text that I hope enriches and clarifies the manuscript.

Journal Requirements:

- The manuscript is now formatted as laid out in the style requirements.

- Supporting Information captions are now at the end of the manuscript file (lines 634-671).

Review Comments to the Author

Reviewer #1: In this manuscript the author compares longevity, locomotor activity, and sleep between three strains of Drosophila melanogaster and between males and females for each strain. The results suggest that, particularly regarding age associated changes in sleep, that strains vary significantly in the extent to which sleep changes over time and that major differences are observed between the sexes within these strains, particularly with regard to changes in sleep as flies age. The study contrasts these strain and sex specific differences in sleep with age associated changes in anticipatory locomotor activity, which appears to be characterized by less strain-to-strain variability. The study offers a useful cautionary reminder that age-related changes in sleep will be highly dependent on the strain and sex of the flies studied. However, there are a few concerns that might be addressed by the author to improve the study.

- I thank the reviewer for their time, their recognition of the potential merits of the manuscript, and their constructive suggestions. I hope the reviewer finds, as I do, that the revised manuscript is more robust in its conclusions and discussion points after the additions and changes that have been suggested.

The major concern here is the apparent absence of replication. For all behavioral experiments a sample size of “17-32” is reported for each strain and sex. This suggests that a single monitor was run for each strain/sex, leaving open the question of how replicable the strain and sex differences described were in the hands of the author. This significantly limits the impact of the study. This is particularly worrying given that, in several instances, the author describes extremely surprising results that do not fit previously published work. The data shown in figures 4B and C are striking examples, in which female Dah and w1118 flies appear to be nearly insomniac. This is in stark contrast to a significant amount of previous work with the w1118 strain. One wonders if something strange happened during the single run with these flies.

- The reviewer has highlighted an important limitation of the original manuscript. To replicate these findings, I have now run additional experiments with twice as many flies for each sex and strain, at 2 weeks and 6 weeks of age. The results of these experiments are now shown in Figures S2-S6, described in the Methods (lines 426-432 and 453-454), and discussed within each section of the Results (lines 113-118, 143-154, 185-188, 208-214, and 228-232). With respect to the reviewer’s noted concern about the low levels of sleep in Dah and w1118 female flies, this pattern did repeat consistently in the replicate experiments, and in opposition to many published studies. Potential reasons for this discrepancy with the previous literature are now discussed in more specificity in lines 297-307. 

- I wish to be transparent here: the number of monitors and incubators required to run the replicate experiments in parallel at the same time have meant that the analysis window for these experiments was only 24 hours because of practical limitations, as opposed to 48 hours used in the initial experiments. This is spelled out in the Methods, lines 412 and 426-432. While this was an unavoidable limitation in this case, the overall similarities in activity patterns, sleep patterns, anticipation indices, and age-related changes (or lack thereof) add what I hope is a level of confidence in the replicability of the major findings of the study. A notable change in the replicate experiments is that evening anticipation, rather than morning anticipation, was the more robust and significant difference across strains and sexes for 6 weeks versus 2 weeks of age; this has led to changes in the text to focus on morning and evening anticipation collectively, rather than morning anticipation exclusively, in lines 130-131 and 284-295.

Should these strain/sex differences survive replication, it is important to note that such strains are evolving independently in the various labs in which they are being reared. The differences described here could be unique to the author’s lab. This should be noted when discussing differences from previously published results. Related to this, it would be good if the author included information about how long ago the three strains were established in the labs that provided the flies used in this study.

- This is an important point; fly strains very likely diverge in their genetic and epigenetic makeup through their many years being reared in different labs. I have now more thoroughly detailed the source and rearing history of each strain in the Methods section (lines 388-400) and have added to the Discussion section to reiterate that these strains will likely have diverged from ‘the same’ strain in other labs (lines 297-307).

The author should more fully describe the differences between this study and previously published sleep data (there are a great number of studies reporting sleep in CS and w1118 strains and much of it looks very different than what is reported here). This is done to some extent, but the author should spell out what is different about the general profiles and amounts of sleep observed.

- This is now described the Discussion section, stating directly that both the original and the replicate experiments show strikingly low levels of sleep (and particularly night sleep), especially for Dah and w1118 females, compared to previous studies (lines 297-303), and providing some potential reasons for the discrepancy (lines 303-307 and 314-326).

The author should more fully describe the existing literature on sexual dimorphism in Drosophila sleep and circadian timekeeping and place the results described in this context.

- I have now included a section on sexual dimorphism in Drosophila activity and sleep patterns, highlighting the sexually dimorphic neuronal and non-neuronal control mechanisms of the daytime sleep ‘siesta’ behaviour as one exemplar behaviour (lines 344-355).

Other Suggestions:

Line 84: It is not clear what “standard physiological conditions” are.

- Good point; this is now spelled out (lines 77-78).

SEM or some other indication of error should be indicated on the averaged activity and sleep plots shown in Figures 2A-F and 4A-F.

- SEM has been added to Figures 2A-F, 4A-F, S2A-F, and S4A-F; as well as all corresponding captions.

The author should reconsider the logic of the grammar used in lines 202 and 220 which appear to suggest that previously published work came AFTER this manuscript.

- These sentences have been simplified and edited for grammar (lines 269-273 and lines 284-287).

The discussion at the top of page 9 is useful, but the author should acknowledge previous work establishing that single beam activity monitoring overestimates fly sleep (when compared to video tracking or multi-beam monitoring).

- This section now acknowledges work from video tracking systems suggesting that single-beam activity monitoring overestimates fly sleep, as well as studies that have directly compared single-beam, multi-beam, and video tracking approaches (lines 314-326).

Line 325: the author states that “standard fly food” was used. There is no such thing. The author acknowledges elsewhere that diet has major effects on aging (and sleep is quite sensitive to it, too). The author should therefore define the media used (both in vials and in capillary tubes).

- This is a good point – each reference to “standard” food instead now specifies SYA food, which is now more explicitly linked to the recipe originally provided in the Methods section. These changes are in lines 385, 406, and 416.

Reviewer #2: The author completed a thorough analysis of age-related changes to sleep in multiple commonly-used Drosophila strains of flies, and in both sexes. The goal of this analysis was to provide a unifying view of potential reasons for discrepancies between studies using different sexes and strains regarding age-related changes to sleep. The author found that sex differences in sleep and age-related changes to sleep were not the same across strains. The author also found that age-related changes to sleep were not the same across strains. This suggests that both sex and strain matter when studying sleep and age-related changes to sleep, and that in future conclusions must be interpreted with caution in cases where multiple strains or both sexes have not been used. This study will prompt increased rigour in the field and represents a very important contribution. I am also glad the authors have found that females have a longer lifespan, this is something that is common yet difficult to publish considering the one study that reported a longer male lifespan.

- Thank you to the reviewer for their insightful comments. With respect to female/male differences in lifespan, on reflection the manuscript could have been more inclusive in its consideration of the many factors that contribute to sexual dimorphism in lifespan. I have now added sentences to discuss this point (lines 259-265).

---

Thank you again for your time and consideration. I look forward to hearing from you. 

Kind regards,

Dr Nathan Woodling

---

## [Decision Letter · Decision Letter 1]

29 Jul 2024

Sex- and strain-dependent effects of ageing on sleep and activity patterns in Drosophila

PONE-D-24-06034R1

Dear Dr. Woodling,

We’re pleased to inform you that your manuscript has been judged scientifically suitable for publication and will be formally accepted for publication once it meets all outstanding technical requirements.

Kind regards,

Efthimios M. C. Skoulakis, PhD

Academic Editor

PLOS ONE

Additional Editor Comments (optional):

Reviewers' comments:

Reviewer's Responses to Questions

**Comments to the Author**

1. If the authors have adequately addressed your comments raised in a previous round of review and you feel that this manuscript is now acceptable for publication, you may indicate that here to bypass the “Comments to the Author” section, enter your conflict of interest statement in the “Confidential to Editor” section, and submit your "Accept" recommendation.

Reviewer #1: All comments have been addressed

2. Is the manuscript technically sound, and do the data support the conclusions?

Reviewer #1: Yes

3. Has the statistical analysis been performed appropriately and rigorously? 

Reviewer #1: Yes

4. Have the authors made all data underlying the findings in their manuscript fully available?

Reviewer #1: Yes

5. Is the manuscript presented in an intelligible fashion and written in standard English?

Reviewer #1: Yes

6. Review Comments to the Author

Reviewer #1: The author has addressed all of my concerns with additional runs of the previously described experiments and has provided important context in the text.

I have no further concerns.

7. PLOS authors have the option to publish the peer review history of their article (what does this mean?). If published, this will include your full peer review and any attached files.

Reviewer #1: No

---

## [Editor Report · Acceptance letter]

8 Aug 2024

PONE-D-24-06034R1 

PLOS ONE

Dear Dr. Woodling, 

I'm pleased to inform you that your manuscript has been deemed suitable for publication in PLOS ONE. Congratulations! Your manuscript is now being handed over to our production team.

Kind regards, 

on behalf of

Dr. Efthimios M. C. Skoulakis 

Academic Editor

PLOS ONE